# Environmental variability supports chimpanzee behavioural diversity

Ammie K. Kalan [ID] et al.[#]

Large brains and behavioural innovation are positively correlated, species-specific traits, associated with the behavioural flexibility animals need for adapting to seasonal and unpredictable habitats. Similar ecological challenges would have been important drivers throughout human evolution. However, studies examining the influence of environmental variability on within-species behavioural diversity are lacking despite the critical assumption that population diversification precedes genetic divergence and speciation. Here, using a dataset of 144 wild chimpanzee (*Pan troglodytes*) communities, we show that chimpanzees exhibit greater behavioural diversity in environments with more variability — in both recent and historical timescales. Notably, distance from Pleistocene forest refugia is associated with the presence of a larger number of behavioural traits, including both tool and non-tool use behaviours. Since more than half of the behaviours investigated are also likely to be cultural, we suggest that environmental variability was a critical evolutionary force promoting the behavioural, as well as cultural diversification of great apes.

---

[#]A list of authors and their affiliations appears at the end of the paper.

Variation in the brain size of many nonhuman primates (hereafter primates) and birds is strongly associated with behavioural flexibility and innovation propensity[1–9]. Moreover, higher rates of behavioural innovation and large brain size correlate with markers of technical, cultural and social intelligence[1,2,4–8,10,11]. Species with larger brains are also found in habitats that are more seasonal and are thus able to survive during times of resource scarcity[1,3,7], for example, nonmigratory birds[2]. Collectively, this research suggests that larger-brained species have greater innovative capacities and are better able to adapt and survive in novel, or more variable environments[1,2,5,7,9].

Although brain size and innovation rate are positively correlated across primates[1,5,6,8], there is limited evidence linking these traits to sources of environmental variation (e.g. seasonality). One study found support for innovation rate being positively associated with climatic variability, including coefficient of variation in precipitation[12], whereas a replication of this study, with a larger number of primate species, failed to confirm the correlation[4]. The authors suggested that their measures of climatic variability failed to capture historical processes that might influence present-day behavioural variation among primates.

Similarly, many human behavioural innovations are thought to have evolved in response to the changing and fluctuating environments endured throughout the Plio-Pleistocene[13–16]. As such, multiple environmental hypotheses have been proposed for hominin evolution. Among these, habitat-specific hypotheses[17–19] stress the importance of moving from ancestral environments, most notably from closed, wet rainforests to more open and arid savannah grasslands[20,21]. Other hypotheses emphasize the degree of climate and habitat unpredictability over time as the primary ecological force promoting behavioural diversification, such as the variability selection hypothesis[20,22]. Nevertheless, it is generally agreed that environmental variability, whether in the short or long-term, shaped the adaptive suite of characteristics found in the genus *Homo*: habitual bipedalism, a large neocortex, tool use and manufacture, cooperative hunting of large game, control of fire, and complex social and cultural cognitive abilities[13,16,18,20,23]. However, the degree to which more open and arid habitats, rather than the severity of ecological fluctuations over time, were a driving force for hominin behavioural diversification is difficult to reconstruct using the limited fossil record. Instead, using a comparative approach[24,25], the influence of environmental variability on behavioural diversity can be tested in nonhuman great apes, for which empirical data are available.

Due to the limited geographic range or the lack of behavioural variation present in a single species, previous studies have tested for the effect of seasonality on behaviour across multiple taxa[4,9]. However, this approach limits inference about external effects due to species-specific variation in intrinsic traits. Therefore by focusing on one of humankind's closest living relatives, chimpanzees, we can test the influence of environmental variability on within-species traits, given that these great apes have a wide geographic range across Equatorial Africa, ranging from forested to savannah woodland habitats, whilst also exhibiting substantial population variation in behaviours[26,27]. These include tool use behaviours for extractive foraging, some of which have also been shown to be cultural (i.e. group-specific social traditions)[28–30]. One previous study examined the effect of annual rainfall on the distribution of cultural behaviours in chimpanzees and found mixed support using cladistic analyses[31]. Building on this research, we use a much larger dataset on chimpanzee behavioural diversity and conduct regression analyses to test multiple sources of ecological and environmental variation.

We find that chimpanzee communities experiencing greater seasonality, living further away from historical Pleistocene forest refugia, and predominantly located in a savannah woodland habitat, are more likely to possess a diverse set of 31 behavioural traits. These results are robust to the categorization of behaviours and do not depend on the inclusion of particular chimpanzee field sites. Overall, seasonal and unstable environmental conditions are associated with greater within-species behavioural diversity, providing comparative support for variability acting as an evolutionary force favouring behavioural and cultural diversification more generally.

## Results

**Environmental variability in the past and present.** We investigated the potential influence of environmental variability on the behavioural diversity of chimpanzee communities, considering both recent and historical ecological-evolutionary processes[32]. We used spatially explicit data to capture environmental variability for each chimpanzee community (or social group) on three different timescales: the short-, mid-, and long-term. First, to capture short-term ecological variability, we used precipitation seasonality (i.e. coefficient of variation in monthly rainfall) averaged across 30 years (1970–2000) extracted from the WorldClim database[33] where larger values represent greater variation in rainfall. Precipitation is a primary factor for assessing ecosystem productivity[34,35]; therefore, it may be considered a proxy for food resource availability. Rainfall has also been linked to multiple life history traits across primate species, including gestation length and lifespan[36]. Important to note here is that precipitation seasonality is not acting as an environmental constraint (i.e. it does not in itself prevent certain behaviours from being expressed by chimpanzees due to the lack of suitable resources being present) but is rather a proxy for short-term ecological conditions (see ref. [31] and Methods for more details).

Second, as a proxy of mid-term variability, we assessed the predominant habitat type as either closed forest (forest) or relatively open savannah woodlands (savannah), assuming that present-day habitats and their ecosystems took longer to establish than a few decades. We considered savannah habitats as more ecologically variable than forested habitats given they have more pronounced seasons and greater fluctuation in climate[37]. Although this dichotomization likely underestimates the full spectrum of environmental variation[38], we constrained the classification to these two categories since the shift from a closed and wet, to an open and arid habitat has been emphasized repeatedly as a critical tipping point for human evolution[19–21]. We classified any community where fragmented, tropical, lowland, wet, or montane forest is present as forest and all others as savannah woodland (see Methods for additional details).

Lastly, to assess long-term effects of environmental variability, we tested the distance to Pleistocene forest refugia (Fig. 1) using Maley's designations for Africa during the last glacial maximum[39]. Pleistocene forest refugia are estimated to have been present from 10,000 to ~2.5 million years ago[40], an epoch marked by repeated forest expansion and contraction during cycles of glaciation, and an overall drier and cooler climate[41]. As such, tropical forest refugia that remained intact throughout the Pleistocene provided a stable climate and habitat for some populations over thousands of years[42]. Therefore, as with other primate taxa[32], the subsequent shifts in vegetation and range limits of forest refugia are expected to have left a historical signal on the distribution and adaptations of species today[41–44]. The stable nature of Pleistocene forest refugia can be described as either cradles or museums of biodiversity. As cradles, refugia act as diversification pumps, as seen in many tropical regions where species richness is high[45,46]. Alternatively, as museums, refugia offer stable environmental conditions that favour the persistence of species over time with low extinction rates[45,47]. Importantly,

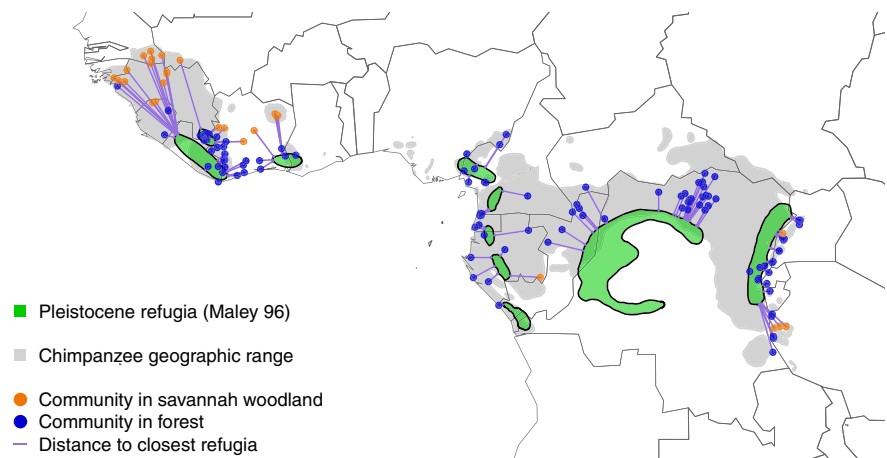

**Fig. 1 Distance of chimpanzee communities to Pleistocene forest refugia.** The green areas depict the Pleistocene forest refugia as described by Maley[39], purple lines show straight-line distances calculated between the center point of a chimpanzee community to the limit of the nearest forest refuge, and dots represent a unique chimpanzee community in a predominantly forest (blue) or savannah woodland (orange) habitat. Chimpanzee geographic range plotted according to the IUCN 2018[77].

the size and duration of refugia, as well as the behavioural ecology of a species, will interact in space and time to create evolutionary cradles or museums of biological diversity[42,48]. Using this framework, we predict that chimpanzee behavioural diversification occurred in one of two ways; under relatively stable conditions, characteristic of forest refugia, or under increasingly variable environments as populations dispersed and moved away from refugia, innovated new behaviours to adapt, and colonized new habitats[42,45,48].

**Chimpanzee behavioural and cultural diversity**. To test whether the behavioural diversity of a chimpanzee community is a function of the degree of environmental variability, either in the past or recent present, we updated a previously published dataset[27] of 144 distinct chimpanzee communities. We coded 31 behaviours for each community (1/0) as either present (direct or indirect evidence) or else not observed[27]. These 31 behaviours are not universal to all chimpanzees; rather they exhibit population-level, including cultural variation[28–30]. Importantly, this dataset includes behaviours that have been shown to be adaptive to local environmental or ecological conditions, such as the use of caves and bathing in savannah chimpanzees to aid thermoregulation during times of heat stress[25,37]. The 31 behaviours also include a number of foraging traits where chimpanzees in some communities learn to extract particular resources, often with the use of tools (e.g. algae fishing[49], ant dipping[50,51], pestle pounding[52], nut cracking[53,54], and tool-assisted hunting[55]). Although the adaptive nature of these behaviours has largely been assumed to be nutritional benefits and dietary flexibility, there is currently no evidence linking these behaviours, directly or by proxy, to reproductive success. We do know that nut cracking provides chimpanzees with a substantial net energy gain[56], and tool-assisted hunting observed in one community of chimpanzees permits individuals with less strength, namely females and young, to capture and consume energetically-rich vertebrate meat[55].

In our analysis, we modeled the probabilistic occurrence of behaviours (out of a possible 31) per community as a function of one of the three environmental variability predictors. We controlled for the expectation that more behaviours are likely to be documented if a community has been observed for longer[27]. Using R (R Core Team 2017, version 3.5.3), we fitted Bayesian Regression Models (BRM) with a Bernoulli response distribution and logit link function. In total, we fitted three BRMs, one with

each predictor, since it was not possible to test all three simultaneously due to collinearity. Additionally, we included in all BRMs as fixed effects the four currently recognized chimpanzee subspecies, number of observation months per community, and the human-footprint value[57] at the center of each community, as this was already found to negatively impact chimpanzee behavioural diversity[27]. Furthermore, we accounted for spatial autocorrelation in each model. We present all results using weak priors and further investigate the influence of priors on the results (see Methods for details; Supplementary Fig. 1).

The probability of occurrence across all behaviours per chimpanzee community was positively affected by all three environmental variability predictors (Table 1; Fig. 2) with the most prominent effects found for distance to Pleistocene forest refugia. Chimpanzee communities located further from historical Pleistocene forest refugia had a higher probability of the 31 behaviours being present (estimate (mean of the marginal posterior distribution) ± sd (standard deviation of the marginal posterior distribution) = 0.523 ± 0.228; 95% credible interval (CI) = [0.072, 0.977]). Chimpanzee behaviours were also more likely to occur in environments with greater precipitation seasonality (i.e. larger coefficients of variation; 0.314 ± 0.255, [−0.199, 0.794]) and in predominantly savannah woodland habitats relative to forested habitats (0.608 ± 0.596, [−0.591, 1.745]; Table 1; Fig. 2), but these effects were less pronounced. The proportion of the posterior distribution supporting a positive association between each environmental variability predictor and chimpanzee behavioural diversity ranged from 99% for Pleistocene forest refugia, 89% for CV precipitation, and 85% for a savannah woodland relative to forested habitat (Table 1).

To assess the robustness of these results, we fitted the models with three alternative response variables for quantifying chimpanzee behavioural diversity: the number of behavioural categories, tool use behaviours, and non-tool use behaviours occurring per community. We constructed 13 behavioural categories according to the resource targeted by foraging behaviours (e.g. ants, termites, and algae) or with respect to its general presumed function (e.g. communication, water extraction, and thermoregulation)[27]. We further classified behaviours as tool use, non-tool use, or otherwise unknown (Supplementary Data 1). For the 13 behavioural categories, all three predictors had considerable influence although effects were again most pronounced for distance to Pleistocene forest refugia. There was a higher probability of finding more behavioural categories in

**Table 1 Results of three Bayesian Regression Models based on weak priors testing the probability of occurrence of all 31 behaviours in a chimpanzee community (N = 144) as a function of three predictors of environmental variability.**

|  |  | Estimate | Sd | CI 2.5% | CI 97.5% | Post. dist. >0 |
|---|---|---|---|---|---|---|
|  | Intercept[a] | −4.410 | 0.499 | −5.428 | −3.463 | – |
| Environmental variability predictors | Distance to refugia | 0.523 | 0.228 | 0.072 | 0.977 | 0.990 |
|  | CV precipitation | 0.314 | 0.255 | −0.199 | 0.794 | 0.888 |
|  | Habitat_savannah | 0.608 | 0.596 | −0.591 | 1.745 | 0.849 |
| Control predictors[a] | Human footprint | −0.300 | 0.161 | −0.630 | 0.004 | 0.031 |
|  | Observation months | 0.905 | 0.294 | 0.327 | 1.491 | 0.998 |
|  | ssp_ellioti | 0.112 | 0.692 | −1.302 | 1.402 | 0.577 |
|  | ssp_schweinfurthii | −0.173 | 0.547 | −1.258 | 0.876 | 0.400 |
|  | ssp_troglodytes | 0.227 | 0.645 | −1.085 | 1.445 | 0.544 |

[a]Average given across all three models.
The mean of the marginal posterior distribution (estimate), standard deviation of the marginal posterior distribution (sd) and the 2.5% and 97.5% credible intervals centred on the mean (CI) and proportion of the posterior distribution greater than zero are given.

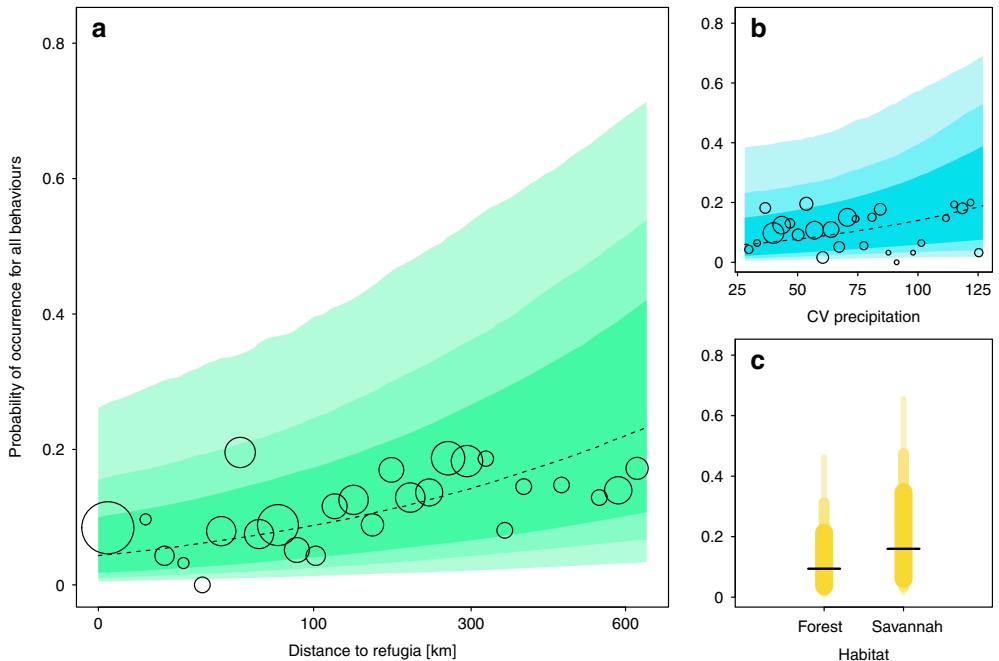

**Fig. 2 The probability of occurrence for 31 chimpanzee behaviours per community as a function of environmental variability.** These behaviours are more likely to occur when chimpanzees live in habitats **a** further away from Pleistocene forest refugia, with **b** greater precipitation seasonality, and **c** a predominantly savannah woodland landscape. The size of the circles in plots **a** and **b** indicates the sample size, or number of chimpanzee communities per value of the predictor where the total n = 144 chimpanzee communities. The coloured areas depict the 67, 87, and 97% credible intervals centred on the mean predicted posterior distribution (**a**, **b** dashed line, **c** horizontal line) for the probability of occurrence across all 31 behaviours.

chimpanzee communities further away from Pleistocene forest refugia (0.473 ± 0.359, [−0.262, 1.156]), and to a lesser extent, in environments with larger variation in precipitation (0.331 ± 0.327, [−0.337, 0.930]), and in savannah woodland habitats (0.710 ± 0.723, [−0.729, 2.070]). The proportion of the posterior distribution supporting a positive association between each environmental predictor and chimpanzee behavioural diversity, measured via behavioural categories, ranged from 95 to 84% (Supplementary Table 1). We found similar effects for distance to Pleistocene refugia on the occurrence of tool use and non-tool use behaviours while the other two environmental predictors had more variable and smaller effects (Fig. 3; Supplementary Table 1). For all models, the control variable of observation months positively affected chimpanzee behavioural occurrence whilst human footprint had a negative impact as already shown in a previous study[27] (Table 1, Supplementary Table 1, Supplementary Fig. 2).

## Discussion

This study supports the general prediction that environmental variability fosters greater behavioural diversity. Specifically, we find evidence for historical, long-term effects of environmental variability, namely distance from Pleistocene forest refugia, to have a more pronounced and consistent influence on the probability of occurrence of 31 chimpanzee behaviours, compared with the mid- or short-term variation in habitat type or precipitation seasonality, respectively. However, both precipitation seasonality and a savannah woodland habitat, relative to a forested one, were also positively associated with chimpanzee behavioural diversity. Although seasonality has been shown to correlate with behavioural innovation and flexibility in cross-species comparisons of birds[1,2,7] and to some extent across primates[12] (but also see ref. [4]), this study shows that environmental variability generally promotes within-species behavioural diversification as well.

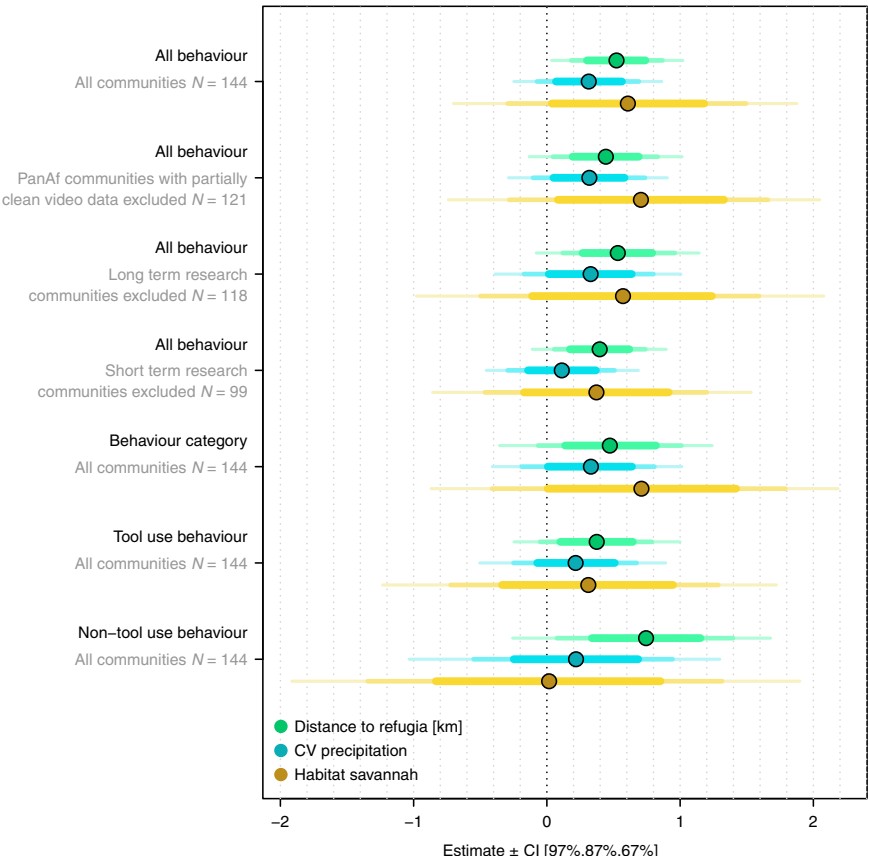

**Fig. 3 Effects of all three environmental variability predictors on chimpanzee behavioural diversity.** We tested precipitation seasonality, savannah woodland versus forest habitat, and distance to Pleistocene refugia on the probability of occurrence for 31 chimpanzee behaviours as well as various subsets of the data to verify robustness of the results. The plot shows the mean of the marginal posterior distribution (dots) and the 67, 87, and 97% credible intervals centred on the mean (coloured areas).

Since the behaviours targeted in this study include group-specific and cultural traits[27–30], we infer that behavioural and cultural diversity of chimpanzees is supported by environmental variation in ecological conditions. Consequently, a large number of behaviours, cultural or otherwise, are found in chimpanzee communities living under conditions that are more seasonally variable in the short-term and historically more unstable and unpredictable environments. There may also be the potential for chimpanzees modifying their habitats via cultural behaviours. Such cultural niche construction processes are characteristic of human societies[58]. Additional data on the role of social learning for the behaviours investigated in this study, and their interactions with the environment, could provide further opportunities for investigating cultural niche construction in chimpanzees.

The positive association between distance from Pleistocene forest refugia and behavioural diversity suggests that populations of *Pan troglodytes* that dispersed and moved away from refugia over time may have been more likely to innovate, and retain (including culturally), additional behaviours that facilitated adaptation to novel habitats. Meanwhile, chimpanzee communities that remained within the more stable forest refugia, may have been more static in their behavioural diversity, a process that potentially reflects the museum hypothesis whereby minimal population differentiation and diversification is predicted within refugia[45–48]. Alternatively, these refugial populations may have suffered a loss of behavioural diversity due to a lack of sustained selection pressures or stochasticity. Both processes could explain why chimpanzee communities

within or close to forest refugia show comparatively little behavioural and cultural diversity.

There is also the possibility that chimpanzee populations throughout the Pleistocene may have survived in the savannah habitats surrounding forest refugia, which would have expanded as refugial forests contracted[42]. These savannah refugia are known to have shaped the biogeography of present day African savannah ungulates[59]. Therefore, it is plausible that select chimpanzee populations, with behavioural adaptations for living in dry savannah woodlands, may have survived repeated glacial and climatic extremes of the Pleistocene in non-forest refugia. Indeed, present-day chimpanzees exhibit behavioural and physiological adaptations to aid survival in both forest and savannah[25,37]. To disentangle these different evolutionary scenarios, we need detailed data on chimpanzee genetic diversity and population history across their geographic range[26]. Only then will we be able to track the migration of populations and test whether dispersal patterns support repeated colonization by chimpanzees originating from Pleistocene forest refugia over chimpanzee populations remaining within savannah refugia and adapting to changing environmental conditions. Previous research on chimpanzees at some regional scales show that genetic differentiation among populations is correlated with environmental variation[60]. However, sampling often restricts broad species-level inferences from such smaller-scale studies[61] and furthermore, genome-behavioural trait associations remain largely unknown for chimpanzees.

Overall, our results suggest that changing or variable environments were more important than climate or habitat stability for supporting chimpanzee behavioural diversity. For example, the majority of the tool use behaviours considered in this study are used for extractive foraging, whereby these behavioural innovations likely conferred a selective advantage[5,6], as in nut cracking[53,56] and tool-assisted hunting[55]. More specifically, chimpanzee populations dispersing from refugia to colonize new environments, or those adapting to changing environments, may have benefited from expanding their dietary breadth to include new or unpredictable food resources. This may explain why environments with more pronounced seasonality (savannah woodland habitats and larger coefficient of variation in precipitation) tend to support more behaviours. Behavioural adaptations for thermoregulation are also particularly useful in drier habitats and are not observed in chimpanzee communities living in forests[25,37]. Such behavioural flexibility has already been linked to invasion success of novel habitats by birds[2,9] and has been suggested to widen ecological tolerance and facilitate niche expansion in chimpanzees[37].

Chimpanzee habitats are dramatically changing as human-modified landscapes become increasingly common across Africa[62]. Such anthropogenic disturbance also affects climatic conditions, exemplified by global warming and extreme local weather fluctuations that are drying out environments[63,64]. This study suggests that chimpanzee behavioural and cultural flexibility may be valuable for responding to climate change and enabling survival in new habitats. The reliance on cultural traits in some animal taxa, such as whales, elephants, and chimpanzees, has recently garnered attention for its urgent consideration into species conservation[27,65,66]. Unfortunately, previous research has shown that chimpanzee behavioural and cultural diversity is currently threatened due to widespread human impact[27]. Therefore, some chimpanzee communities may struggle to survive in the near future as behaviours are lost due to increasing fragmentation caused by human activities and shifting ranges due to greater aridification[63,65,67,68].

We find that the distribution of chimpanzee behavioural diversity has been clearly affected by both historical and recent sources of environmental variability. If we assume that our last common ancestor with nonhuman primates was similar to present-day chimpanzees, then this suggests that they too may have used a diverse toolbox of behaviours to adapt and survive. Since behaviour is difficult to reconstruct from fossils alone, a comparative framework using nonhuman primates for insight into human evolution can facilitate empirical research. For example, baboons (*Papio* sp.) have been proposed as a model for human evolution because of their continental-wide species radiations as well as behavioural and ecological specializations[69,70]. However, it is difficult to separate genetically inherited predispositions from flexible adaptations when comparing across species. Therefore, the present study provides a new perspective into intraspecific behavioural diversity using chimpanzees, demonstrating that fluctuating environments over space and time may have been important drivers of behavioural and cultural diversification, in addition to speciation, for other great apes, including humans[15,20].

In summary, the evolution of chimpanzee behavioural diversity and flexibility across their range was likely influenced by a combination of adapting to novel environments while being constrained by multiple historical effects (e.g. genetic, behavioural, and physiological), from past climate and habitat changes experienced by previous populations. Therefore, focusing on a single catalyst for human evolution is likely trivial since environmental variability, over both recent and historical timescales, may have had compound consequences on extinct and living hominins, as is evident for chimpanzee behavioural diversity observed today.

## Methods

**Behaviour data**. We used a behavioural dataset compiled for 144 chimpanzee communities already used in a previous publication[27] and updated here with additional information (Supplementary Data 1). The data for 46 of these communities come from the Pan African Programme: the Cultured Chimpanzee (hereafter PanAf) while we compiled data for another 106 (eight of which also include PanAf data) from the published literature. PanAf data were collected in the field for a minimum of one full year whenever possible (observation period 12–30 months for 37 chimpanzee communities; 1–10 months for 9 communities), using a systematic grid design of 1 by 1 km cells. which covered an area that varied in size depending on the habitat and topography of the landscape (9–143 km$^2$). Since chimpanzees were unhabituated to human presence during PanAf data collection, we primarily used remote, infrared sensor camera traps (Bushnell Trophy cameras) to collect behavioural observations as well as a uniform data collection protocol applied at all PanAf sites (for full details of the methods applied in the field, see http://panafrican.eva.mpg.de/english/approaches_and_methods. php). We installed at least one camera-trap per grid cell and complemented this with additional camera trapping, targeting locations regularly visited by wild chimpanzees such as animal trails, natural log bridges, fruiting trees, and tool-use sites. For this study, we selected behaviours that were detectable using our PanAf methodology, i.e. could be observed on camera traps and/or indirectly via observable traces that are left behind in the faeces (e.g. feeding remains) or artefacts (e.g. discarded tools). Additionally, we selected behaviours that show variation across chimpanzee populations, with many of the 31 behaviours also qualifying as cultural[28–30] in previous studies (Supplementary Data 1).

For the remaining 106 chimpanzee communities, we extracted behavioural data from the published literature by screening both printed and electronic articles, books, and dissertations. We used Google Scholar to search for publications by pairing chimpanzee with key words such as "tool" and "tool-use", as well as the 31 different behaviours chosen for the study. We would also identify additional publications by screening the reference lists of articles and books already compiled. In total, we screened approximately 450 primary resources published from 1951 to 2017, with the majority published after the 1980s (see Supplementary Data 1 for full list of reference materials).

The behavioural dataset comprised 31 possible combinations for all 144 chimpanzee communities with the occurrence (1/0) of a particular behaviour coded within it. We coded presence (1) whenever direct observations (via camera-trap or human observer) or indirect observations (artefacts, feeding remains, and faecal samples) provided evidence for a particular chimpanzee community engaging in a particular behaviour. Whenever no evidence was documented or found, we coded the behaviour as not observed (0). We do not use the term absent, as these do not reflect true absences since observations of behaviours are contingent upon observation effort, which is a critical control in the analysis (see below). This is also why our analysis uses a probabilistic approach. Moreover, with respect to environmental constraints, if there were data reported to suggest that particular resources targeted by a given behaviour were not available within the territory of a given chimpanzee community, we coded this as NA (not applicable). In total, we coded 31 behaviours for each community including foraging for insects, algae, honey, or nuts, extracting water, thermoregulation and communication. We broadly categorized all behaviours into one of 13 categories and further defined them as tool use, non-tool use behaviour or tool use unknown (e.g. when termites or ants were found only in the dung). See also ref. [27] and Supplementary Data 1.

**Environmental variability data**. We used three predictors of environmental variability to capture its effects on three different timescales: short-, mid,- and long-term variation. For short-term variability, we extracted precipitation seasonality from the derived bioclimatic variable (BIO15) of the WorldClim database (https://www.worldclim.org/) version 1.4 with a 1 km$^2$ spatial resolution. This value was calculated by averaging monthly coefficients of variation in precipitation between 1960 and 1990, with a minimum input of 10 years per location[33].

To capture mid-term environmental variability, we classified chimpanzee communities as predominantly savannah woodland or forest by overlaying classifications based on freely available GIS layers (WWF Terrestrial-Ecoregions of the World: https://www.worldwildlife.org/publications/terrestrial-ecoregions-of-the-world; The Nature Conservancy's Terrestrial-Ecoregions and Biomes of the World: http://www.landscope.org/map_descriptions/ecosystems/tnc_ecoregional_boundaries/15602/; and the European Commission's Global Landcover Data for Africa: https://forobs.jrc.ec.europa.eu/products/glc2000/products.php). For the majority of communities, the separation between forest and savannah woodland habitats was clear. The forest category included any site where fragmented, lowland, montane, tropical or humid forest was present according to the layers. For 11 of the 144 chimpanzee communities the consensus across layers suggested a mosaic between savannah woodland and forested habitat. Here, we referred to descriptions of the habitat used by chimpanzees in the literature, or via PanAf field observations, to assign the area as being predominantly savannah woodland or forest (Supplementary Data 1).

To capture effects of environmental variability on a longer time scale, we used the distance of present-day chimpanzee communities from the nearest Pleistocene forest refugia as designated by Maley[39] since his map offers the greatest spatial resolution for African forest refugia. More specifically, we calculated in kilometers the straight-line distances to the limit of the nearest Pleistocene refuge[39] from the center coordinates of each chimpanzee community (Supplementary Data 1; Fig. 1). The distance was zero if a community was located inside a designated forest refuge.

**Statistical analysis**. To analyze the impact of environmental variability on the observed behavioural diversity in chimpanzee communities, we used Bayesian Regression Models (BRMs) with Bernoulli response distribution and logit link function. As the response variable, we used four different measures to account for behavioural diversity. First, we used the occurrence (1/0) of a behaviour per community; second, we categorized the behaviours into 13 categories and used the occurrence of a category per community; third, we considered only the occurrence of tool use behaviours, and fourth, we used only the occurrence of non-tool use behaviours.

The three environmental variability predictors were highly correlated (Pearson correlation coefficients between distance to refugia and coefficient of variation in precipitation ($rP = 0.720$, $N = 144$, $P < 0.001$), distance to refugia and habitat type ($rP = 0.660$, $N = 144$, $P < 0.001$), and coefficient of variation in precipitation and habitat type ($rP = 0.613$, $N = 144$, $P < 0.001$). Therefore, we fit three different models comprising one predictor at a time. Additionally, we included in each model, as control effects, the human-footprint value for each community based on the coordinates at its center[57,71], the number of months the community was observed, and the currently recognized chimpanzee subspecies (*Pan troglodytes verus/ellioti/schweinfurthii/troglodytes*). As random effects, we included the site and behaviour into the model. Additionally, we included as random slopes the environmental predictor, the human footprint, the number of months of observation within the site, and the chimpanzee subspecies within behaviour, as well as the correlation parameters between the random intercepts and random slope terms[72,73]. To control for spatial autocorrelation we included a Gaussian process over the longitude and latitude for each community[74] by using the function *gp* from the R package 'brms'[75]. This revealed that the spatial covariance among communities declined towards zero after a 50–70 km distance (Supplementary Table 2; Supplementary Fig. 4). Prior to fitting the models, we checked all predictors for their distribution and, consequently, square-root transformed distance to refugia to achieve a more normal distribution. After this we z-transformed all numerical predictors to a mean of zero and a standard deviation of one[73]. Since we fit Binomial and Bernoulli response distributions, we did not transform the control variable observation time. However, we did check the effect of observation time if it had been log transformed, as is commonly performed for Poisson distributed models, and found negligible change in the results (Supplementary Fig. 3). See Supplementary Code 1 for the complete specification of the full models.

We fitted all models in R (R Core Team 2017, version 3.5.3) using the function *brm* from the R- package 'brms' (version 2.8.0)[75], which runs by default 2000 iterations over four MCMC chains, including a warm-up period of 1000 iterations per chain resulting in 8000 usable posterior samples[75]. We are confident in the accuracy of the MCMC results because: (1) visual inspection showed stationarity and convergence to a common target, (2) all Rhat[76] values were below 1.01, and (3) there were no divergent transitions after warm-up. We used the default flat priors and in addition, we tested all models with a weak and wide prior for all predictors and control variables. As a weak prior, we chose a normal distribution with a mean of 0 and a standard deviation of 1 and for a wide predictor we used a mean of 0 and a standard deviation of 10. Since environmental variability was predicted to positively influence, and favour behavioural diversification, we hypothesized it would positively affect the occurrence probability of chimpanzee behaviours and therefore report the percent of the posterior distribution that supports a positive association between the predictors and chimpanzee behavioural diversity. Additionally, we assessed support for each predictor via the distribution of credible intervals centred on the mean estimate of the marginal posterior distribution.

We verified the robustness of our results by removing PanAf sites where video data were not yet fully cleaned at the time of data analyses (23 communities lost). We also verified that results did not hinge upon the inclusion of long-term research sites, which includes all chimpanzee communities that have been habituated to observers, by excluding them and rerunning models (26 communities lost), and also by excluding all communities with the lowest observation effort of 1 month (45 communities lost). In all cases, model estimates did not vary considerably other than for non-tool use behaviours, which were underrepresented in this study (Fig. 3).

**Reporting summary**. Further information on research design is available in the Nature Research Reporting Summary linked to this article.

## Data availability
The data used for this paper are available in Supplementary Data 1. The publicly available datasets used in this study are available at the following links: WCS, CIESIN and Columbia University Last of the Wild Project, Global Human Footprint Dataset v.2, https://doi.org/10.7927/H4M61H5F; https://sedac.ciesin.columbia.edu/data/set/wildareas-v2-human-footprint-geographic; WorldClim database, https://www.worldclim.org/; WWF Terrestrial-Ecoregions of the World, https://www.worldwildlife.org/publications/terrestrial-ecoregions-of-the-world; Nature Conservancy's Terrestrial-Ecoregions and Biomes of the World, http://www.landscope.org/map_descriptions/ecosystems/tnc_ecoregional_boundaries/15602/; European Commision's Global Landcover Data for Africa, https://forobs.jrc.ec.europa.eu/products/glc2000/products.php.

## Code availability
The R code used to run the analyses for this study is available in the file Supplementary Code 1.

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

## Acknowledgements
This research was generously funded by the Max Planck Society, Max Planck Society's Innovation Fund, and the Heinz L. Krekeler Foundation. We thank the following authorities for kindly granting permission to conduct research in their respective countries: Ministère de la Recherche Scientifique et de l'Innovation, Ministère des Forêts et de la Faune, and the Conservation Society of Mbe Mountains, Cameroon; Ministère de la Recherche Scientifique, Ministère des Eaux et Forêts and Ministère de l'Environnement, Côte d'Ivoire; Institut Congolais pour la Conservation de la Nature and Ministère de la Recherche Scientifique, Democratic Republic of Congo; Agence Nationale des Parcs Nationaux, Centre National de la Recherche Scientifique et Technologique and Société Equatoriale d'Exploitation Forestière, Gabon; Department of Wildlife and Range Management and the Forestry Commission, Ghana; Ministère de l'Agriculture de l'Elevage et des Eaux et Forêts, Guinea; Instituto da Biodiversidade e das Áreas Protegidas and Ministro da Agricultura e Desenvolvimento Rural, Guinea-Bissau; Forestry Development Authority, Liberia; Ministre de l'Environnement et de l'Assainissement et du Developpement Durable and des Eaux et Forêts, Mali; Conservation Society of Mbe Mountains and National Park Service, Nigeria; Ministère de l'Economie Forestière, Ministère de le Recherche Scientifique et Technologique and Agence Congolaise de la Faune et des Aires Protégées, Republic of Congo; Ministry of Education and Rwanda Development Board, Rwanda; Direction des Eaux, Forêts Chasses and La Conservation des Sols, and Réserve Naturelle Communautaire de Díndéfélo, Senegal; Ministry of Agriculture, Forestry and Food Security and the National Protected Area Authority, Sierra Leone; Tanzania Commission for Science and Technology and Tanzania Wildlife Research Institute, Tanzania; Uganda National Council for Science and Technology, Uganda Wildlife Authority and Makerere University Biological Field Station, Uganda. We would also like to extend special thanks to several collaborators who facilitated or helped with PanAf

research at various field sites including: Arcel Bamba, Donatienne Barubiyo, Matthieu Bonnet, Gita Chelluri, Chloe Chipoletta, Katherine Corogenes, Jean Claude Dengui, Theophile Desarmeaux, Karsten Dierks, Emmanuel Dilambaka, Andrew Dunn, Villard Ebot Egbe, Henk Eshuis, Marcel Ketchen Eyong, David Fine, Theo Freeman, John Hart, Martijn Ter Heegde, Thurston Cleveland Hicks, Inaoyom Imong, Michael Kaiser, Mbangi Kambere, Laura Kehoe, Vincent Lapeyre, Joshua M. Linder, Nuria Maldonado, Giovanna Maretti, Rumen Martin, Michael Masozera, Tanyi Julius Mbi, Vianet Mihindou, Yasmin Moebius, Geoffrey Muhanguzi, Felix Mulindahabi, Mizuki Murai, Protais Niyigaba, Nadege Wangue Njomen, Nicolas Ntare, Abel Nzeheke, Robinson Orume, Bruno Perodeau, Jill Pruetz, Sebastien Regnaut, Emilien Terrade, Alexander Tickle, Els Ton, Joost van Schijndel, Lilah Sciaky, Alhaji Malikie Siaka, Paul Telfer, Richard Tshombe, Hilde Vanleeuwe, Elleni Vendras, and Kyle Yurkiw.

## Author contributions

A.K.K., L.K., and H.S.K. designed the study; M.A., C.B., P.D., M.M., and H.S.K. managed data collection; A.K.K., L.K., M.A., P.D., C.D.B., C.B., and F.H. compiled data for this study; A.K.K., P.D., A.A., S.A., F.A., E.A.A., E.B., D.B., M.B., G.B., V.E.B., H.C., K.C., C. Co., R.C., K.D., E.Di., V.E.E., J.M.F., A-C.G., J.H., D.H., V.H., T.C.H., S.J., J.J., P.K., M.K., Mo.K., I.K., J.L., B.L., K.L., V.Le., M.L., G.M., S.M., R.M., T.J.M., A.C.M., D.M., F.M., M.M., S.N., P.N., E.No., L.J.O., J.P., A.R., S.R., C.S., C.T., A.T., E.T., J.v.S., E.V., A.W., E.G.W., J.W., Y.G.Y., and K.Y. collected data in the field; E.E.A., M.Bo., R.C., C.C., B.C., E.D., T.D., D.D., A.D., A.G., I.I., K.J.J., D.K., K.E.L., V.L., E-N.M., B.M., D.M., E.N., R.O., L.P., A.P., M.R., A.R., C.S., V.S., F.S., N.T., H.V., V.V., R.M.W., and K.Z. provided infrastructure and logistical support for data collection; A.K.K., L.K., and H.S.K. analyzed the data; A.K.K. drafted the paper with feedback from all co-authors who also approved the final draft.

## Funding

## Competing interests

The authors declare no competing interests.

## Additional information

Ammie K. Kalan [1✉], Lars Kulik[1], Mimi Arandjelovic [1], Christophe Boesch [1,2], Fabian Haas[1], Paula Dieguez [1], Christopher D. Barratt[1,3], Ekwoge E. Abwe[4,5], Anthony Agbor [1], Samuel Angedakin[1], Floris Aubert[2], Emmanuel Ayuk Ayimisin[1], Emma Bailey[1], Mattia Bessone[1], Gregory Brazzola[1], Valentine Ebua Buh[1], Rebecca Chancellor[6,7], Heather Cohen[1], Charlotte Coupland[1], Bryan Curran[8], Emmanuel Danquah[9], Tobias Deschner[1], Dervla Dowd[2], Manasseh Eno-Nku[10], J. Michael Fay[11], Annemarie Goedmakers [12], Anne-Céline Granjon[1], Josephine Head[1], Daniela Hedwig[13], Veerle Hermans[14], Kathryn J. Jeffery [15,16], Sorrel Jones [1], Jessica Junker[1,3], Parag Kadam [17], Mohamed Kambi[1], Ivonne Kienast[1], Deo Kujirakwinja[8], Kevin E. Langergraber[18], Juan Lapuente[1,19], Bradley Larson[1], Kevin C. Lee [1,18], Vera Leinert[2], Manuel Llana [20], Sergio Marrocoli[1], Amelia C. Meier[1], Bethan Morgan[4,5,15], David Morgan[21,22], Emily Neil [1,23], Sonia Nicholl [1], Emmanuelle Normand[2], Lucy Jayne Ormsby[1], Liliana Pacheco [20], Alex Piel[24,25], Jodie Preece[1], Martha M. Robbins[1], Aaron Rundus[6], Crickette Sanz [22,26,27], Volker Sommer[25,28], Fiona Stewart [24], Nikki Tagg [14,23], Claudio Tennie[29], Virginie Vergnes[2], Adam Welsh[1], Erin G. Wessling [1,30], Jacob Willie[14], Roman M. Wittig [1,31], Yisa Ginath Yuh [1], Klaus Zuberbühler[32,33] & Hjalmar S. Kühl [1,3]

[1]Max Planck Institute for Evolutionary Anthropology, 04103 Leipzig, Germany. [2]Wild Chimpanzee Foundation, 04103 Leipzig, Germany. [3]German Centre for Integrative Biodiversity Research (iDiv) Halle-Jena-Leipzig, Leipzig 04103, Germany. [4]Ebo Forest Research Project, BP3055 Messa, Cameroon. [5]Institute for Conservation Research, San Diego Zoo Global, Escondido, CA 92027-7000, USA. [6]West Chester University, Department of Psychology, West Chester, PA 19382, USA. [7]West Chester University, Department of Anthropology and Sociology, West Chester, PA 19382, USA. [8]Wildlife Conservation Society, New York, NY 10460, USA. [9]Department of Wildlife and Range Management, Faculty of Renewable Natural Resources, Kwame Nkrumah University of Science and Technology, Kumasi, Ghana. [10]WWF Cameroon Country Programme Office, BP 6776 Yaoundé, Cameroon. [11]Wonga-Wongue Reserve, Libreville, Gabon. [12]Chimbo Foundation, 1011 PW Amsterdam, Netherlands. [13]Elephant Listening Project, Bioacoustics Research Program, Cornell Lab of Ornithology, Cornell University, Ithaca, NY 14850, USA. [14]KMDA, Centre for Research and Conservation, Royal Zoological Society of Antwerp, B-2018 Antwerp, Belgium. [15]School of Natural Sciences, University of Stirling, Stirling, UK. [16]Agence National des Parcs Nationaux, BP20379 Libreville, Gabon. [17]University of Cambridge, Cambridge, UK CB2 3QG.

[18]School of Human Evolution and Social Change & Institute of Human Origins, Arizona State University, Tempe, AZ 85287, USA. [19]Comoé Chimpanzee Conservation Project, Würzburg, Germany. [20]Jane Goodall Institute Spain and Senegal, Dindefelo Biological Station, Dindefelo, Kedougou, Senegal. [21]Lester E. Fisher Center for the Study and Conservation of Apes, Lincoln Park Zoo, Chicago, IL 60614, USA. [22]Wildlife Conservation Society, Congo Program, B.P. 14537 Brazzaville, Republic of Congo. [23]Born Free Foundation, Broadlands Business Campus, Horsham RH12 4QP, UK. [24]School of Natural Sciences and Psychology, Liverpool John Moores University, James Parsons Building, Liverpool L3 3AF, UK. [25]University College London, Department of Anthropology, London WC1H 0BW, UK. [26]Washington University in Saint Louis, Department of Anthropology, St. Louis, MO 63130, USA. [27]Kyoto University Institute for Advanced Study, Kyoto University, Yoshida-Ushinomiya-cho, Sakyo-Ku, Kyoto 606-8501, Japan. [28]Gashaka Primate Project, Serti, Taraba, Nigeria. [29]Department for Early Prehistory and Quaternary Ecology, University of Tübingen, 72070 Tübingen, Germany. [30]Department of Human Evolutionary Biology, Harvard University, Cambridge, MA 02138, USA. [31]Taï Chimpanzee Project, CSRS, Abidjan, Côte d'Ivoire. [32]Université de Neuchâtel, Institut de Biologie, 2000 Neuchâtel, Switzerland. [33]School of Psychology and Neuroscience, University of St Andrews, St Andrews, UK. ✉email: ammie_kalan@eva.mpg.de

