## [Peer Review File · Nature Communications]

Reviewers' Comments:

Reviewer #1:

Remarks to the Author:

Kalan et al.'s paper, Environmental variability supports chimpanzee behavioural diversity uses an extensive data set several field sites of 31 chimpanzee behaviors (from camera traps and previous published data) and environmental data to understand linkages between them. Generally, the authors found that this suite of data supports the conclusion that environmental variability fosters greater behavioral diversity in chimpanzees. Based on this observation, the authors propose two alternative scenarios to explain the cultural diversity of chimpanzees in forest versus dry savanna locations. The strongest predictor for a lack of behavioral diversity was distance from Pleistocene refuges, leading them to suggest that chimpanzees in rainforests may have lost behavioral diversity due to a lack of selection in a variable environment, while those that migrated out following post-glacial forest expansion would have experienced more variable environments and thus acquired additional behaviors as they colonized new areas. Alternatively, they propose that some chimpanzee populations might have persisted in dry forest environments during periods of maximum glaciation, and their persistence in the highly variable environments may have facilitated behavioral diversity.

Strengths:

The paper presents the most comprehensive comparison of chimpanzee behaviors across their range combined with extensive environmental data (which have been previously published).

The data are analyzed using an impressive array of regression analyses, and definitely convinced me that there are strong positive correlations between environmental variation and behavioral variation.

They present two bold hypotheses alternative hypotheses about the diversity of chimpanzee behaviors in rainforest versus dry savanna habitats and link them back to hominin evolution.

The analyses presented are robust and well-justified.

This paper will be of very broad interest.

Weaknesses:

This paper up until the discussion is excellent, but the authors need to scale back the discussion in revision for several reasons:

1. Lack of evidence that the behaviors included in the study impact fitness: The authors assume that these cultural behaviors are traits that passed down through generations and that these behaviors impact fitness in the environments where they are found. In order to do this, they need to provide evidence that these 31 traits are the products of social learning (which they do for a few) and also that these traits impact fitness. Without evidence that the traits included in the study impact fitness, there is no reason to conclude that increased behavioral diversity is adaptive in chimpanzees in different environments. The authors try to circumvent this issue by pointing out that a suite of traits in the genus Homo (bipedalism, large neocortex, cooperative hunting, etc.) is widely considered to be adaptive.
2. The paper does not present novel data, only previously published data interpreted in a new way: The underlying data have already been published in Science. This paper is a reanalysis and interpretation of those data. Without evidence that these behaviors are under selection or at least linked with variation in genotypes among chimpanzees in different environments, the paper is (very unfortunately) plagued by a lot of hand waving and is only novel in that presents two hypotheses that

can be neither supported nor refuted in this paper.

3. The lack of genetic data limits placing the data into an evolutionary context: A true test of these hypotheses needs to include genotype data analyzed alongside the behavioral data and various environmental predictors

The distinction between short-, mid- and long- term variation is unnecessarily confusing. It does not add much to the paper to have these distinctions. The authors should consider just stating that three main predictor variables were used: precipitation seasonality, habitat type and distance from Pleistocene refuges. There is really no reason to have these classified into different time scales at the outset of the and would be better explained in the discussion.

There are other papers that have linked genetic variation and environmental variation in chimpanzees, e.g., papers by Mitchell et al. (Environmental variation and rivers govern the structure of chimpanzee genetic diversity in a biodiversity hotspot. BMC Evolutionary Biology (DOI 10.1186/s12862-014-0274-0); Male-driven differences in chimpanzee community structure is tied to habitat variation in Cameroon and Nigeria. International Journal of Primatology, published online August 10, 2018. (<https://doi.org/10.1007/s10764-018-0053-7>.) The authors should consider incorporating these into the discussion to strengthen the argument that there is a clear link between environmental variation and genetic variation in chimpanzees. It would help strengthen the foundations of the hypotheses presented in the discussion.

Summary of recommendation

The concepts presented in this paper are excellent. However, the underlying data have already been published previously leaving really the hypotheses as the novel contributions of this manuscript. In light of the fact that the behavioral data are not clearly linked to fitness, much of the discussion is simply speculation and not a "smoking gun." If the paper is selected for a revision and resubmission, the authors should consider providing stronger evidence that the suite of behaviors is under selection, and if possible, include genotype data in the analysis. If that is not possible, then perhaps the authors could simply expand on the importance of environmental variability in promoting behavioral diversity in chimpanzees and minimize trying to link the observed behavioral variation to environmentally-driven natural selection.

Reviewer #2:

Remarks to the Author:

This is an incredibly interesting and well-executed study that tests whether behavioral flexibility, measured as the size of the behavioral repertoire, is associated with environmental variability *within* a species, chimpanzees. This provides an important test of the hypothesis that environmental unpredictability selects for increased behavioral flexibility, which is one of the dominant account of the evolution of human culture and cognitive skills. The authors have assembled an impressive dataset including a large number of chimpanzee populations (the n of which should probably be mentioned in the abstract), and find that three different measures of environmental variability - distance to glacial refugia, savanna habitat, and rainfall variability - all predict greater behavioral diversity. This supports the hypothesis and implies that similar selection pressures could have acted during human evolution.

I think this will be an important paper that should be of interest to a wide audience including

behavioral ecologists, primatologists, anthropologists, and other evolutionary human scientists. The paper is clearly and concisely written, and I specifically appreciate the transparent reporting of the statistical methods and the availability of code and data. My only larger concern, and it is by no means a fatal one, is with the choice of the priors, as I will detail below. Dealing with this does not require a major revision of the paper, only some slight adjustments to the methods and results, and will not change the major conclusions. I therefore strongly recommend acceptance of this paper pending the necessary revisions outlined below.

Priors: the authors carefully report the priors chosen (thank you!) and did a sensitivity analysis (Supplemental Figure 1) to show how the choice of priors affects the results. This is excellent protocol, and I also appreciate the journal encouraging (enforcing?) this, as evidenced by the accompanying check sheet. My concern is that the choice of the informative priors is circular, and should in fact not be done in this case. The authors (lines 433-440) argue that (i) informative priors were chosen for the environmental variability predictors because these were "assumed" to be positively associated with behavioral repertoire, and (ii) informative priors were chosen for the human footprint variable based on their previous study.

My issue is that in (i) having a hypothesis about this association is NOT the same as having actual prior results (from a different study); indeed, this is circular as the authors hypothesize a positive association, formalize this hypothesis with the priors having a positive mean (and pretty tight SD), and then continue to find a positive association. In fact, the parameter estimates for the three environmental variability measures are all quite close to 0.4 (ranging from 0.38 to 0.48), which was the prior mean, and it thus seems as if the models merely returned the prior! At this point the skeptical reader has to go to supplementary figure 1 to see that the models using flat or weak priors (centered on 0) actually return similar estimates, so there was in fact enough information in the data to arrive at these estimates and they weren't simply returning the prior. I believe that this is highly problematic, as many readers will not take this extra step and may get a false impression based on the circular informative priors. I therefore think it is necessary to remove the results with the informative priors and instead report the results with the weak priors in the paper (I don't think flat priors are interesting; weak priors can regularize the model and are therefore to be preferred). Indeed, I believe a more proper way to deal with the fact that the authors had a clear hypothesis would be to report the % of the posterior that supports a positive association between environmental variability and behavioral repertoire (I suspect it would be around 90%) - this will give a direct measure of our confidence in the association given these data. Furthermore, this would be a truly probabilistic inference, which the authors claim to do in line 361 but then their inference nevertheless relies on binary thresholds based on the (arbitrary) 95% credible intervals. Why not report the % of the posterior supporting the predicted association instead of or alongside the 95%CI?

The issue with (ii) is similar, as the results from the authors' previous study on human footprint were based at least partly *on the same data* and therefore cannot necessarily be treated as prior information. Now arguably, since the previous study was only based on a third or so of the populations in the current study, these previous associations could be treated as pilot data, but in the very least this requires some more justification to avoid raising the circularity flag in skeptical readers.

Other minor issues:

358-360: One might also have considered a zero-inflation model, wherein the probability of the absence of a behavior reflecting a true 0 (behavior is indeed absent) or a false 0 (behavior is in fact present but was not recorded) is modeled, ideally as a function of observation effort. Please explain the difference to the approach used (perhaps in supplement), justify why it is preferable over a zero-inflation approach, and ideally also do a comparison to an ZI model (which is easy to do in brms).

408: Three separate models were fit because the predictors were correlated. This is fine, but one might also have thought of combining the predictors using some form of factor analysis - briefly justify why three separate models were preferred

415-417: Excellent that the analyses controlled for spatial autocorrelation using Gaussian processes! indeed, it would be really interesting to know what the Gaussian process results look like! Please add minimally the relevant parameter estimates to the supplement and briefly summarize in the main text (e.g. are close communities similar in their behavioral repertoires and how quickly does this similarity decay with distance?), and ideally it would be nice to illustrate these processes with a figure in the supplement (see e.g. McElreath 2016 Statistical Rethinking Figures 13.9 and 13.10).

Reviewer #3:

Remarks to the Author:

The authors tested association between environmental variability and behavioural diversity in chimpanzee communities. The study is well set up except that there is a critical issue in the statistical analysis, that is the authors set informative prior on the regression coefficients between environmental variability and behavioural diversity and the posterior distribution of these regression coefficients is similar to the priors. This means that either the prior agrees with the data or the data is not informative and the prior drives the result. One way to test this is to use different priors and see if the posterior stays the same. A standard way for regression under Bayesian framework is to use a normal distribution with mean zero and large variance as the prior for regression coefficient. This is what I suggest the authors to do, and I cannot give recommendations on the decision on the paper until the authors can convince me that their result is not driven by the prior they chosen.

Reviewers' comments:

Reviewer #1 (Remarks to the Author):

Kalan et al.'s paper, Environmental variability supports chimpanzee behavioural diversity uses an extensive data set several field sites of 31 chimpanzee behaviors (from camera traps and previous published data) and environmental data to understand linkages between them. Generally, the authors found that this suite of data supports the conclusion that environmental variability fosters greater behavioral diversity in chimpanzees. Based on this observation, the authors propose two alternative scenarios to explain the cultural diversity of chimpanzees in forest versus dry savanna locations. The strongest predictor for a lack of behavioral diversity was distance from Pleistocene refuges, leading them to suggest that chimpanzees in rainforests may have lost behavioral diversity due to a lack of selection in a variable environment, while those that migrated out following post-glacial forest expansion would have experienced more variable environments and thus acquired additional behaviors as they colonized new areas. Alternatively, they propose that some chimpanzee populations might have persisted in dry forest environments during periods of maximum glaciation, and their persistence in the highly variable environments may have facilitated behavioral diversity.

Strengths:

The paper presents the most comprehensive comparison of chimpanzee behaviors across their range combined with extensive environmental data (which have been previously published).

The data are analyzed using an impressive array of regression analyses, and definitely convinced me that there are strong positive correlations between environmental variation and behavioral variation.

They present two bold hypotheses alternative hypotheses about the diversity of chimpanzee behaviors in rainforest versus dry savanna habitats and link them back to hominin evolution.

The analyses presented are robust and well-justified.

This paper will be of very broad interest.

We thank the reviewer for taking the time to provide such a thorough peer review and the generally positive appraisal of our paper.

Weaknesses:

This paper up until the discussion is excellent, but the authors need to scale back the discussion in revision for several reasons:

1. Lack of evidence that the behaviors included in the study impact fitness: The authors assume that these cultural behaviors are traits that passed down through generations and that these behaviors impact fitness in the environments where they are found. In order to do this, they need to provide

evidence that these 31 traits are the products of social learning (which they do for a few) and also that these traits impact fitness. Without evidence that the traits included in the study impact fitness, there is no reason to conclude that increased behavioral diversity is adaptive in chimpanzees in different environments. The authors try to circumvent this issue by pointing out that a suite of traits in the genus Homo (bipedalism, large neocortex, cooperative hunting, etc.) is widely considered to be adaptive.

This is a good point and one we fully acknowledge is difficult to address given the need for longitudinal reproductive and survival data that would be necessary to compare across wild chimpanzee populations, with or without particular behaviours, to evaluate fitness (note, these data are not even available for the few long-term chimpanzee field sites under study).

To address the reviewers concern, we now include a paragraph where we cite population-specific studies on chimpanzees where researchers have found support for certain chimpanzee behaviours to be adaptive to local environmental and ecological conditions. These examples primarily include thermoregulation and foraging behaviours which are included in the 31 chimpanzee behaviours in our study (lines 175-184; Page 7).

2. The paper does not present novel data, only previously published data interpreted in a new way: The underlying data have already been published in Science. This paper is a reanalysis and interpretation of those data. Without evidence that these behaviors are under selection or at least linked with variation in genotypes among chimpanzees in different environments, the paper is (very unfortunately) plagued by a lot of hand waving and is only novel in that presents two hypotheses that can be neither supported nor refuted in this paper.

Yes, the behavioural data have been published previously but we do not agree with the reviewer that this should be a reason to devalue the impact of our paper. Rather, under the current need for greater transparency and openness in science, we would hope that the journal and reviewers would appreciate that we have made all our data freely accessible to the public in order to promote open sharing of our knowledge in the hopes that others will continue to build on it. Moreover, as is common in many areas of science, especially genetics, datasets are usually published alongside original papers and then re-evaluated and re-analyzed to address additional hypotheses and these studies are still worthy of high-impact science. We do not see why our large dataset on chimpanzee behaviours, which is a first of its kind, should be treated differently.

The aim of this study was to specifically investigate within species behavioural diversity, not genetic diversity (see abstract lines for reasoning 67-69). We are also still a long way off from understanding the complex genome-trait association in chimpanzees, not to mention even our own species. Furthermore, we expect some, and possibly all behaviours investigated, to be inherited through mechanisms of cultural evolution and not necessarily reflected in genetic diversity (see relevant discussion points lines 242-250 where we also removed specific mention of 'selection'). For these reasons, using the behavioural dataset is a valid approach to investigating large-scale patterns of environmental variation on chimpanzee behavioural diversity across Africa.

3. The lack of genetic data limits placing the data into an evolutionary context: A true test of these hypotheses needs to include genotype data analyzed alongside the behavioral data and various environmental predictors

Ideally, adding genetic data would be wonderful but we simply do not have genotype data from these 144 chimpanzee communities. More importantly however, genotype data would still be insufficient for our purposes since there is no clear reason to suggest that, for example, genetic diversity (e.g., Fst; a commonly used measure in genetic studies) is associated with particular behavioural phenotypes or their diversity in chimpanzees. We would also like to point out that we did include subspecies as a control variable in our mixed models, which some might consider as an important control for potential genetic structuring within chimpanzees.

Therefore we would argue that the value of genetic variation as a proxy for selection would nonetheless fall short in significant ways with respect to the goals of this study: 1) Measures of genetic differentiation do not tell us whether particular behaviours are present or absent, and 2) the 31 chimpanzee behaviours we investigated are complex traits, most of which are socially learned and cultural, and if genetic in basis, are expected to be caused by polygenic and epigenetic effects. To truly evaluate selection of specific behaviours therefore requires higher resolution genomic data including testing for positive selection rather than rough measures of genetic data, which is beyond the scope of this work and will likely take generations of scientists to achieve.

Importantly, as mentioned earlier, there are no genetic samples or data available from all 144 chimpanzee communities and we do not think this will ever be possible considering that for some of these communities only a few individuals remain in the wild. We feel a nuanced discussion of why our manuscript focusses solely on behavioural data and does not include genetics would detract from the overall aims of our paper. However, we added to the discussion an explanation of what is known from genetic studies (thanks to the reviewer's recommendations) and also why genetics, at this time, is still limited with respect to this study's goals (page 12, lines 273-276).

The distinction between short-, mid- and long- term variation is unnecessarily confusing. It does not add much to the paper to have these distinctions. The authors should consider just stating that three main predictor variables were used: precipitation seasonality, habitat type and distance from Pleistocene refuges. There is really no reason to have these classified into different time scales at the outset of the and would be better explained in the discussion.

We find the distinction between recent and historical timescales of environmental variation to be integral to our study since previous research has only focused on one but not both, and previous research has explicitly mentioned that including historical effects would be important to consider (reference #4). Therefore, short, mid and long-term environmental variability is a helpful trichotomy to explain the variation in time-depth captured by each variable. We hope the reviewer can

appreciate this reasoning. We did however limit their use now to only the first descriptions of the environmental predictors to reduce confusion.

There are other papers that have linked genetic variation and environmental variation in chimpanzees, e.g., papers by Mitchell et al. (Environmental variation and rivers govern the structure of chimpanzee genetic diversity in a biodiversity hotspot. BMC Evolutionary Biology (DOI 10.1186/s12862-014-0274-0); Male-driven differences in chimpanzee community structure is tied to habitat variation in Cameroon and Nigeria. International Journal of Primatology, published online August 10, 2018. (<https://doi.org/10.1007/s10764-018-0053-7>)). The authors should consider incorporating these into the discussion to strengthen the argument that there is a clear link between environmental variation and genetic variation in chimpanzees. It would help strengthen the foundations of the hypotheses presented in the discussion.

These are great, thank you for bringing them to our attention. We now cite the most relevant article (the first one) regarding environmental variation associated with genetic differentiation among chimpanzee populations (ref 58) and another paper published in the same issue by the same lab in another part of the manuscript's discussion with respect to potential effects of climate change (ref 66; lines 273-276).

Summary of recommendation

The concepts presented in this paper are excellent. However, the underlying data have already been published previously leaving really the hypotheses as the novel contributions of this manuscript. In light of the fact that the behavioral data are not clearly linked to fitness, much of the discussion is simply speculation and not a “smoking gun.” If the paper is selected for a revision and resubmission, the authors should consider providing stronger evidence that the suite of behaviors is under selection, and if possible, include genotype data in the analysis. If that is not possible, then perhaps the authors could simply expand on the importance of environmental variability in promoting behavioral diversity in chimpanzees and minimize trying to link the observed behavioral variation to environmentally-driven natural selection.

As mentioned above, we do not have (nor know of anyone that does have) genotype data for all 144 chimpanzee communities, therefore following the reviewer's concerns and advice we now provide more information regarding the adaptive nature of the behaviours we included in this study (lines 175-184) as evidence for their presumed fitness advantage, and we also qualified and edited our interpretation in the discussion with regards to selection (removed twice from discussion; only appears now once on line 258).

Reviewer #2 (Remarks to the Author):

This is an incredibly interesting and well-executed study that tests whether behavioral flexibility,

measured as the size of the behavioral repertoire, is associated with environmental variability *within* a species, chimpanzees. This provides an important test of the hypothesis that environmental unpredictability selects for increased behavioral flexibility, which is one of the dominant account of the evolution of human culture and cognitive skills. The authors have assembled an impressive dataset including a large number of chimpanzee populations (the n of which should probably be mentioned in the abstract), and find that three different measures of environmental variability - distance to glacial refugia, savanna habitat, and rainfall variability - all predict greater behavioral diversity. This supports the hypothesis and implies that similar selection pressures could have acted during human evolution.

Thank you for the excellent summary and positive review of the manuscript. The sample size of 144 chimpanzee communities has now been added to the abstract (line 69).

I think this will be an important paper that should be of interest to a wide audience including behavioral ecologists, primatologists, anthropologists, and other evolutionary human scientists. The paper is clearly and concisely written, and I specifically appreciate the transparent reporting of the statistical methods and the availability of code and data. My only larger concern, and it is by no means a fatal one, is with the choice of the priors, as I will detail below. Dealing with this does not require a major revision of the paper, only some slight adjustments to the methods and results, and will not change the major conclusions. I therefore strongly recommend acceptance of this paper pending the necessary revisions outlined below.

Thank you very much for your positive recommendation and your helpful concerns regarding the priors. We have adjusted the paper accordingly and detail the changes following the points below.

Priors: the authors carefully report the priors chosen (thank you!) and did a sensitivity analysis (Supplemental Figure 1) to show how the choice of priors affects the results. This is excellent protocol, and I also appreciate the journal encouraging (enforcing?) this, as evidenced by the accompanying check sheet. My concern is that the choice of the informative priors is circular, and should in fact not be done in this case. The authors (lines 433-440) argue that (i) informative priors were chosen for the environmental variability predictors because these were "assumed" to be positively associated with behavioral repertoire, and (ii) informative priors were chosen for the human footprint variable based on their previous study.

My issue is that in (i) having a hypothesis about this association is NOT the same as having actual prior results (from a different study); indeed, this is circular as the authors hypothesize a positive association, formalize this hypothesis with the priors having a positive mean (and pretty tight SD), and then continue to find a positive association. In fact, the parameter estimates for the three environmental variability measures are all quite close to 0.4 (ranging from 0.38 to 0.48), which was the prior mean, and it thus seems as if the models merely returned the prior! At this point the skeptical reader has to go to supplementary figure 1 to see that the models using flat or weak priors (centered on 0) actually return similar estimates, so there was in fact enough information in the data to arrive at these estimates and they weren't simply returning the prior. I believe that this is highly problematic, as many readers will not

take this extra step and may get a false impression based on the circular informative priors. I therefore think it is necessary to remove the results with the informative priors and instead report the results with the weak priors in the paper (I don't think flat priors are interesting; weak priors can regularize the model and are therefore to be preferred).

The reviewer makes an excellent point and indeed we were conflating the hypothesis we had regarding environmental variability and behavioural repertoires with the choice of informative priors. We now report the main figures and results table with the weak prior, as suggested (see new Figure 2 & 3 and Table 1 and completely updated figures and tables for the Supplementary). The results have also been updated to report the estimate and standard deviation of the mean of the marginal posterior distribution using the weak prior in the main text (lines 199-206 & lines 218-221). Moreover, our prior sensitivity analysis now includes weak, flat and wide priors, with informative priors no longer being used (SI figure 1).

Indeed, I believe a more proper way to deal with the fact that the authors had a clear hypothesis would be to report the % of the posterior that supports a positive association between environmental variability and behavioral repertoire (I suspect it would be around 90%) - this will give a direct measure of our confidence in the association given these data. Furthermore, this would be a truly probabilistic inference, which the authors claim to do in line 361 but then their inference nevertheless relies on binary thresholds based on the (arbitrary) 95% credible intervals. Why not report the % of the posterior supporting the predicted association instead of or alongside the 95%CI?

This is a good point, and we opted to now add the % of the posterior that supports a positive association (our hypothesis) into the revised Table 1 as well as adding this to the revised writing of the results (lines 206-209 & 221-224) and in Supplementary Table 1.

The issue with (ii) is similar, as the results from the authors' previous study on human footprint were based at least partly *on the same data* and therefore cannot necessarily be treated as prior information. Now arguably, since the previous study was only based on a third or so of the populations in the current study, these previous associations could be treated as pilot data, but in the very least this requires some more justification to avoid raising the circularity flag in skeptical readers.

We now report the results for the human footprint variable and the other control predictors also using weak priors rather than informative, as the reviewer has recommended. The results using weak priors can be found in the revised Table 1, revised Figures 2-3 and revised Supplementary Table 1, revised Supplementary Figure 1 -Supplementary Figure 3.

Other minor issues:

358-360: One might also have considered a zero-inflation model, wherein the probability of the absence of a behavior reflecting a true 0 (behavior is indeed absent) or a false 0 (behavior is in fact present but was not recorded) is modeled, ideally as a function of observation effort. Please explain the difference

to the approach used (perhaps in supplement), justify why it is preferable over a zero-inflation approach, and ideally also do a comparison to an ZI model (which is easy to do in brms).

We did originally think about running a zero-inflated (ZI) poisson model but were actually advised by Richard McElreath (author of reference 72) and others to use a normal binomial (Bernoulli) model. Upon getting this review we thought about the reviewer's suggestion, which we understood was to run a zero-inflated model for a binary response. We were not sure whether this is a valid approach for our data but did give it a try, however we ran into problems fitting such a model. We contacted the developer of the brms package and he explained that it does not make sense to do a ZI with a binary response and suggested that a model with Bernoulli response distribution is sufficient to account for the number of zeros given that observation effort is a critical predictor in the model.

We also reconsidered our original approach to run a ZI poisson model where the response would be counts of behaviours per community but realized that with this approach we actually decrease the number of zeros and we would not capture the probability per behaviour. With regards to the suggestion to fit the ZI as a function of observation effort, we think that observation time is definitely the main source of zeros, therefore it is a critical control variable in the models. Collectively, these reasons led us to choose the current approach as the most appropriate for our data and question.

408: Three separate models were fit because the predictors were correlated. This is fine, but one might also have thought of combining the predictors using some form of factor analysis - briefly justify why three separate models were preferred

Separate models were preferred over a combined single response because 1) CV precipitation is a commonly used seasonality variable in previous studies, many of which we refer to in the introduction, and therefore we found it helpful from a comparative perspective to demonstrate the specific effects of this variable on its own, and 2) we used a habitat type variable in order to especially address the habitat-specific hypotheses (e.g., 'savanna hypothesis'; see lines 96-98) that have been put forth as a driver of hominin evolution and diversification. We further felt the three variables naturally capture different timescales (lines 127-129) and therefore by keeping them separate this can aid understanding the relative effects of each timescale to variation in our model response. We hope the reviewer can appreciate this reasoning.

415-417: Excellent that the analyses controlled for spatial autocorrelation using Gaussian processes! indeed, it would be really interesting to know what the Gaussian process results look like! Please add minimally the relevant parameter estimates to the supplement and briefly summarize in the main text (e.g. are close communities similar in their behavioral repertoires and how quickly does this similarity decay with distance?), and ideally it would be nice to illustrate these processes with a figure in the supplement (see e.g. McElreath 2016 Statistical Rethinking Figures 13.9 and 13.10).

We now added a new supplementary table (SI table 2) showing the estimates and a new supplementary figure (SI Figure 4) which has two panels demonstrating the results of the Gaussian

process and how the spatial covariance is distributed in our data. The left panel plots the communities over the lat long coordinates. The size of the dots corresponds to the number of behaviours found in the communities while the thickness of the lines between communities depicts the degree of covariance between them. The right panel depicts the decay of spatial covariance as a function of distance between communities. It shows that it declines almost to zero after a spatial radius, or distance, of 50-70km. A brief summary has also been added to the methods (lines 393-394).

Reviewer #3 (Remarks to the Author):

The authors tested association between environmental variability and behavioural diversity in chimpanzee communities. The study is well set up except that there is a critical issue in the statistical analysis, that is the authors set informative prior on the regression coefficients between environmental variability and behavioural diversity and the posterior distribution of these regression coefficients is similar to the priors. This means that either the prior agrees with the data or the data is not informative and the prior drives the result. One way to test this is to use different priors and see if the posterior stays the same. A standard way for regression under Bayesian framework is to use a normal distribution with mean zero and large variance as the prior for regression coefficient. This is what I suggest the authors to do, and I cannot give recommendations on the decision on the paper until the authors can convince me that their result is not driven by the prior they chosen.

We thank the reviewer for taking the time to review the statistics and we understand why they were hesitant to comment further. Reviewer 2 had similar concerns and we now report all results using the weak prior which predicts a normal distribution (mean of 0 and sd of 1). See Figure 2, 3 and Table 1 and the Supplementary Information for updated results, as well as lines 197-229 for the updated results in the main text. Please note that in the revised Supplementary Figure 1 that a flat prior and wide prior (mean of 0 and sd of 10) return similar estimates. Thanks to the valuable feedback, we have now removed the informative prior from all methods and results and understand why this was in fact problematic.

Reviewers' Comments:

Reviewer #1:

Remarks to the Author:

This is my second opportunity to review Kalan et al's manuscript.

Strengths

I carefully read through the manuscript and found the authors addressed nearly all of my comments from the last version. I have little to ask of them in revising this version.

The discussion has greatly improved, with more defensible conclusions that are more likely to stand the test of time and more evidence.

It is a really interesting paper, and worthy of publication in Nature or the Nature family of journals.

Weaknesses

Overall, the discussion is more clear and the conclusions are more defensible based on the data presented. My only lingering concern is that the authors be completely up front about the limitations of the study: the behavioral data are not clearly linked to fitness, which is really essential for proving that chimpanzees have experienced environmentally-driven local adaptation. The authors addressed this concern in a few places in the revision and that they modified to lines 175 – 179 to:

“Importantly, this dataset includes behaviours that have been shown to be adaptive to local environmental or ecological conditions, such as the use of caves and bathing in savannah chimpanzees to aid thermoregulation during times of heat stress. The 31 behaviours also include a number of foraging traits where chimpanzees in some communities learn to extract particular resources, often with the use of tools (e.g., algae fishing, ant dipping, pestle pounding, nut cracking, and tool-assisted hunting).”

The reality is that none of the behaviors are linked to variation in reproductive success, either directly or by proxy – possibly with the exception of thermoregulation in savannah chimpanzees (which convincingly showed evidence for adaptive variation). The authors need to be honest and state somewhere near lines 175-179 that none of the behaviors are actually linked to differences in fitness, and only variation in thermoregulation has been linked to adaptive genetic variation. Linking gene-by-environment interactions is central to testing the hypothesis that the environment drives adaptive variation in chimpanzees. The behavioral data presented may or may not test this hypothesis, but taken overall, the study presents quite compelling evidence that the environment has been important in shaping chimpanzee behaviors and probably plays a role in local adaptive variation.

Reviewer #2:

Remarks to the Author:

The authors have carefully and convincingly addressed all my concerns and I no longer have any issues with the paper. I recommend publication as is and congratulate the authors on this highly interesting and impactful study!

Minor issues:

In line 412, I would think that "hypothesized" is a better word than "assumed" since you go on to test these hypotheses?

In the figure legend to supplementary figure 4 (which I am very glad the authors added!) there's an unnecessary ? in the first line. Please also check for typos elsewhere

Reviewer #3:

Remarks to the Author:

The authors have sufficiently solved my previous concern. I have also read through their responses to the other reviewers. I agree with the other reviewers' comments and find that the authors have given satisfactory responses to these comments too. So I'm happy with the manuscript in its current form and thus recommend accepting it.

Point by Point Response to Reviewer Requests

REVIEWERS' COMMENTS:

Reviewer #1 (Remarks to the Author):

This is my second opportunity to review Kalan et al's manuscript.

Strengths

I carefully read through the manuscript and found the authors addressed nearly all of my comments from the last version. I have little to ask of them in revising this version.

The discussion has greatly improved, with more defensible conclusions that are more likely to stand the test of time and more evidence.

It is a really interesting paper, and worthy of publication in Nature or the Nature family of journals.

Thank you very much for your time and effort into reading the revised manuscript and the favourable recommendation.

Weaknesses

Overall, the discussion is more clear and the conclusions are more defensible based on the data presented. My only lingering concern is that the authors be completely up front about the limitations of the study: the behavioral data are not clearly linked to fitness, which is really essential for proving that chimpanzees have experienced environmentally-driven local adaptation. The authors addressed this concern in a few places in the revision was that they modified to lines 175 – 179 to:

“Importantly, this dataset includes behaviours that have been shown to be adaptive to local environmental or ecological conditions, such as the use of caves and bathing in savannah chimpanzees to aid thermoregulation during times of heat stress. The 31 behaviours also include a number of foraging traits where chimpanzees in some communities learn to extract particular resources, often with the use of tools (e.g., algae fishing, ant dipping, pestle pounding, nut cracking, and tool-assisted hunting).”

The reality is that none of the behaviors are linked to variation in reproductive success, either directly or by proxy – possibly with the exception of thermoregulation in savannah chimpanzees (which convincingly showed evidence for adaptive variation). The authors need to be honest and state somewhere near lines 175-179 that none of the behaviors are actually linked to differences in fitness, and only variation in thermoregulation has been linked to adaptive genetic variation. Linking gene-by-environment interactions is central to testing the hypothesis that the environment drive adaptive variation in chimpanzees. The behavioral data presented may or may not test this hypothesis, but taken overall, the study presents quite compelling evidence that the environment has been important shaping chimpanzee behaviors and probably plays a role in local adaptive variation.

We understand the reviewer's concern and agree that it would be a good idea to explicitly state this in the manuscript to avoid any confusion by readers. We added a line explicitly stating that “.....” on page 7, lines 188-190.

Reviewer #2 (Remarks to the Author):

The authors have carefully and convincingly addressed all my concerns and I no longer have any issues

with the paper. I recommend publication as is and congratulate the authors on this highly interesting and impactful study!

Thank you so much for your very helpful review and positive appraisal of our study.

Minor issues:

In line 412, I would think that "hypothesized" is a better word than "assumed" since you go on to test these hypotheses?

Changed to hypothesized.

In the figure legend to supplementary figure 4 (which I am very glad the authors added!) there's an unnecessary ? in the first line. Please also check for typos elsewhere

Removed the '?' and we also checked the entire manuscript for typos.

Reviewer #3 (Remarks to the Author):

The authors have sufficiently solved my previous concern. I have also read through their responses to the other reviewers. I agree with the other reviewers' comments and find that the authors have given satisfactory responses to these comments too. So I'm happy with the manuscript in its current form and thus recommend accepting it.

Thank you very much for taking the time to review our study and for the positive recommendation.